# Improved Adversarial Image Captioning

**Pierre Dognin**[*], **Igor Melnyk**[*], **Youssef Mroueh**[*] , **Jerret Ross**[*], **& Tom Sercu**[*]
IBM Research, Yorktown Heights, NY
{pdognin,mroueh,rossja}@us.ibm.com, {igor.melnyk,tom.sercu1}@ibm.com

## Abstract

In this paper we study image captioning as a conditional GAN training, proposing both a context-aware LSTM captioner and co-attentive discriminator, which enforces semantic alignment between images and captions. We investigate the viability of two discrete GAN training methods: Self-critical Sequence Training (SCST) and Gumbel Straight-Through (ST) and demonstrate that SCST shows more stable gradient behavior and improved results over Gumbel ST.

## 1 Introduction [1]

Significant progress has been made on the task of generating image descriptions using neural image captioning. Early systems were traditionally trained using cross-entropy (CE) loss minimization (Karpathy & Li, 2015; Xu et al., 2015). Later, reinforcement learning techniques (Ranzato et al., 2015; Liu et al., 2017) based on policy gradient methods were introduced to directly optimize metrics such as CIDEr or SPICE Anderson et al. (2016). Along a similar idea, Rennie et al. (2017) introduced Self-critical Sequence Training (SCST), a light-weight variant of REINFORCE, which produced state of the art image captioning results using CIDEr as an optimization metric. To address the problem of sentence diversity and naturalness, image captioning has been explored in the framework of GANs. However, due to the discrete nature of text generation, GAN training remains challenging and has been generally tackled either with reinforcement learning techniques (Hjelm et al., 2017; Rajeswar et al., 2017; Dai et al., 2017) or by using Gumbel softmax relaxation (Jang et al., 2016), as in (Shetty et al., 2017; Kusner & Hernández-Lobato, 2016).

Despite impressive advances, image captioning is far from being a solved task. It remains a challenge to satisfactorily bridge the semantic gap between image and captions to produce diverse, creative, and "human-like" captions. Although applying GANs to image captioning for promoting human-like captions is a very promising direction, the discrete nature of the text generation process makes it challenging to train such systems. The recent work of Caccia et al. (2018) showed that the task of text generation for current discrete GAN models is difficult, often producing unsatisfactory results, and requires therefore new approaches and methods.

In this paper, we propose a novel GAN-based framework for image captioning that enables better language composition, more accurate compositional alignment of image and text, and light-weight efficient training of discrete sequence GAN based on SCST.

## 2 Adversarial Caption Generation

In this Section we present our novel captioner and discriminator models. We employ SCST for discrete GAN optimization and compare it to the approach based on the Gumbel relaxation trick.

**Context Aware Captioner** $G_\theta$. For caption generation we use an LSTM with visual attention (Xu et al., 2015; Rennie et al., 2017) together with a visual sentinel Lu et al. (2017) to give the LSTM a choice to attend to visual or textual cues. While Lu et al. (2017) feeds at each step $t$ only an average image feature, we feed a mixture of image and visual sentinel features from $t-1$ to make the LSTM aware of the last attentional context (called *Context Aware* attention), as seen in Fig. 1. This simple modification gives significant gains, as the captioner is now aware of the visual information used in the past. As reported in Tab. 1, a captioner with an adaptive visual sentinel Lu et al. (2017) gives 99.7 CIDEr versus 103.3 for our Context Aware Attention on COCO validation set.

---

[1]The main part of this work has been accepted by and is planned to be included in the CVPR'19 conference
[*] Alphabetical order; Equal contribution

| Attention Model | CE | RL |
|---|---|---|
| Att2All Rennie et al. (2017) | 98.5 | 115.7 |
| Sentinel Lu et al. (2017) | 99.7 | |
| Context Aware (ours) | 103.3 | 118.6 |

**Table 1:** Performance of captioning systems given various attention mechanisms, Att2All Rennie et al. (2017), sentinel attention Dai et al. (2017) and Context Aware attention on COCO validation set. Models are built using cross-entropy (CE) and SCST Rennie et al. (2017) (RL). Context aware attention brings large gains in CIDEr for both CE and RL trained models.

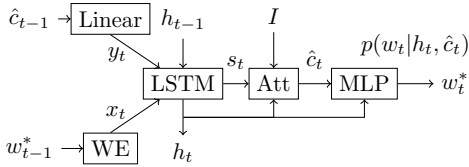

**Figure 1:** Context Aware Captioner. At each step the LSTM is fed with the textual information and a mixture of image features and visual sentinel from the previous steps to make the LSTM aware of the past attentional context.

**Co-attention Pooling Discriminator** $D_\eta$. Previous works jointly embed the modalities at the similarity computation level, referred to as Joint-Emb (e.g., Dai et al. (2017)). Instead, we propose to jointly embed image and caption in earlier stages using a co-attention model Lu et al. (2016) and compute similarity on the attentive pooled representation. We call it a *Co-attention* discriminator, see Fig. 2. In Section 3 we compare $D_\eta$ with Joint-Emb of (Dai et al., 2017; Shetty et al., 2017), where $E_I$ is the average spatial pooling of CNN features and $E_S$ the last state of LSTM.

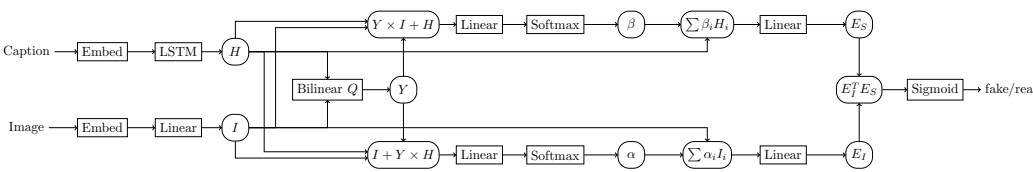

**Figure 2:** Proposed co-attention discriminator (Co-att) architecture. By jointly embedding image and caption with a co-attention model, the discriminator has the ability to modulate the image features depending on the caption and vice versa.

$$G_\theta'^s \rightarrow w^s \rightarrow D_\eta \rightarrow r(w^s) = \log\left(D_\eta\left(I,\left(w_1^s,\ldots,w_T^s\right)\right)\right)$$
$$I$$
$$\boxed{\left(r(w^s) - r(\hat{w})\right)\nabla_\theta \log p_\theta(w^s)}$$
$$G_\theta^* \rightarrow \hat{w} \rightarrow D_\eta \rightarrow r(\hat{w}) = \log\left(D_\eta\left(I,\left(\hat{w}_1,\ldots,\hat{w}_T\right)\right)\right)$$

**Figure 3:** SCST Training of GAN-captioning.

**Training** $D_\eta$. Our discriminator $D_\eta$ is not only trained to distinguish real captions from fake (generated), but also to detect when images are coupled with random unrelated real sentences, thus forcing it to check sentence composition and semantic relationship between image and caption. We solve the following optimization problem: $\max_\eta \mathcal{L}_D(\eta)$, where the loss $\mathcal{L}_D(\eta)$ is

$$\mathbb{E}_{I,w \in S(I)} \log D_\eta(I,w) + \frac{1}{2}\mathbb{E}_{I,w^s \sim p_\theta(.|I)} \log\left(1 - D_\eta(I,w^s)\right) + \frac{1}{2}\mathbb{E}_{I,w' \notin S(I)} \log\left(1 - D_\eta(I,w')\right),$$

where $w$ is the real sentence, $w^s$ is sampled from generator $G_\theta$ (fake caption), and $w'$ is real but random caption.

**Training** $G_\theta$. The generator is optimized to solve $\max_\theta \mathcal{L}_G(\theta)$, where $\mathcal{L}_G(\theta) = \mathbb{E}_I \log D_\eta(I, G_\theta(I))$. The main difficulty is the discrete, non-differentiables nature of the problem. We propose to solve this issue by adopting SCST Rennie et al. (2017) and compare it to the Gumbel relaxation approach of Jang et al. (2016).

**Training** $G_\theta$ **using SCST**. SCST is a REINFORCE variant that uses the reward under the decoding algorithm as baseline. In this work, the decoding algorithm is a "greedy max", selecting at each step the most probable word from $\arg\max p_\theta(.|h_t)$. For a given image, a single sample $w^s$ of the generator is used to estimate the full sequence reward, $\mathcal{L}_G^I(\theta) = \log(D(I,w^s))$ where $w^s \sim p_\theta(.|I)$. Using SCST, the gradient is estimated as follows:

$$\nabla_\theta \mathcal{L}_G^I(\theta) \approx (\log D_\eta(I,w^s) - \underbrace{\log D_\eta(I,\hat{w})}_{\text{Baseline}})\nabla_\theta \log p_\theta(w^s|I) = \left(\log \frac{D_\eta(I,w^s)}{D_\eta(I,\hat{w})}\right)\nabla_\theta \log p_\theta(w^s|I),$$

where $\hat{w}$ is obtained using *greedy max* (see Fig. 3). Note that the baseline does not change the expectation of the gradient but reduces the variance of the estimate. Also, observe that the GAN training can be regularized with any NLP metric $r_{\text{NLP}}$ (such as CIDEr) to enforce closeness of the generated captions to the provided ground truth on the $n$-gram level; the gradient then becomes:

$$\left( \log \frac{D_\eta(I, w^s)}{D_\eta(I, \hat{w})} + \lambda \left( r_{\text{NLP}}(w^s) - r_{\text{NLP}}(\hat{w}) \right) \right) \nabla_\theta \log p_\theta(w^s | I).$$

There are two main advantages of SCST over other policy gradient methods used in the sequential GAN context: **1)** The reward in SCST can be global at the sentence level and the training still succeeds. In other policy gradient methods, e.g., Dai et al. (2017); Liu et al. (2017), the reward needs to be defined at each word generation with the *full* sentence sampling, so that the discriminator needs to be evaluated $T$ times (sentence length). **2)** In (Dai et al., 2017; Liu et al., 2017; Hjelm et al., 2017), many Monte-Carlo rollouts are needed to reduce variance of gradients, requiring many forward-passes through the generator. In contrast, due to a strong baseline, only a single sample estimate is enough in SCST.

**Training $G_\theta$ using the Gumbel Trick.** An alternative way to deal with the discreteness of the generator is by using Gumbel re-parameterization Jang et al. (2016). Define the soft samples $y_t^j$, for $t = 1, \ldots T$ (sentence length) and $j = 1, \ldots K$ (vocabulary size) such that: $y_t^j = \text{Softmax} \left( \frac{1}{\tau} (\text{logits}_\theta(j | h_t, I) + g_j) \right)$, where $g_j$ are samples from Gumbel distribution, $\tau$ is a temperature parameter. We experiment with Gumbel Soft and Gumbel Straight-Through (Gumbel ST) approach, recently used in (Shetty et al., 2017; Kusner & Hernández-Lobato, 2016).

For *Gumbel soft*, we use the soft samples $y_t$ as LSTM input $w_{t+1}^s$ at the next time step and in $D_\eta$, i.e., $\nabla_\theta \mathcal{L}_G^I(\theta) = \nabla_\theta \log(D_\eta(I, y_{1:T}))$. For *Gumbel ST*, we define one-hot encodings $\mathscr{O}_t = \text{OneHot}(\arg\max_j y_t^j)$ and approximate the gradients $\partial \mathscr{O}_t^j / \partial y_t^{j'} = \delta_{jj'}$. To sample from $G_\theta$ we use the hard $\mathscr{O}_t$ as LSTM input $w_{t+1}^s$ at the next time step and in $D_\eta$, hence the gradient becomes $\nabla_\theta \mathcal{L}_G^I(\theta) = \nabla_\theta \log(D_\eta(I, \mathscr{O}_{1:T}))$ Observe that this loss can be additionally regularized with Feature Matching (FM):

$$\mathcal{L}_G^I(\theta) = \log(D_\eta(I, y_{1:T})) - \lambda_F^I \left( ||E_I(w_{1:T}^*) - E_I(y_{1:T})||^2 \right) - \lambda_F^S \left( ||E_{w_{1:T}^*}(I) - E_{y_{1:T}}(I)||^2 \right),$$

where $(w_{1:T}^*)$ is the ground truth caption corresponding to image $I$, and $E_I$ and $E_S$ are co-attention image and sentence embeddings (as defined earlier). Feature matching enables us to incorporate more granular information from discriminator representations of the ground truth caption, similar to how SCST reward can be regularized with CIDEr.

## 3 EXPERIMENTS

**Experimental Setup.** We evaluate our proposed method and the baselines on COCO dataset Lin et al. (2014). Each image is encoded by a resnet-101 He et al. (2016), followed by a spatial adaptive max-pooling to ensure a fixed size of 14×14×2048. An attention mask is produced over the 14×14 spatial locations, resulting in a spatially averaged 2048-dimension representation. LSTM hidden state, image, word, and attention embedding dimensions are fixed to 512 for all models. Before the GAN training, all the models are first pretrained with the cross entropy (CE) loss.

**Experimental Results.** Tab. 2 presents results on COCO dataset for context-aware captioner, two discriminator architectures (ours Co-att, and baseline Joint-Emb) and all training algorithms (SCST, Gumbel ST, and Gumbel Soft). For reference, we also include results for CE (trained only with cross entropy) and CIDEr-RL (pretrained with CE, followed by SCST to optimize CIDEr), as well as results from non-attentional models.

As expected, CIDEr-RL greatly improves the language metrics as compared to CE model (101.6 to 116.1 CIDEr), leading to a significant drop in the vocabulary coverage (from 9.2% to 5.1%). On the other hand, the underperformance of GANs over CIDEr-RL in terms of CIDEr is also expected since GAN's objective is to make the sentences more descriptive and human-like, deviating from the vanilla ground truth captions. The results also show the advantage of our Co-att architecture as compared to the Joint-Emb one, showing the importance of the early joint embedding of the image/caption pair for better similarity computation. Regularizing GANs with CIDEr additionally

**Table 2:** Performance of the models mentioned in this work on COCO dataset. The results are averaged from 4 independent training runs.

| | CIDEr | | BLEU4 | | ROUGEL | | METEOR | | Vocab. cover | |
|---|---|---|---|---|---|---|---|---|---|---|
| CE | 101.6 | ±0.4 | 0.312 | ±.001 | 0.542 | ±.001 | 0.260 | ±.001 | 9.2 | ±0.1 |
| CIDEr-RL | **116.1** | ±0.2 | **0.350** | ±.003 | **0.562** | ±.001 | 0.269 | ±.000 | 5.1 | ±0.1 |
| $GAN_1$(SCST, Co-att, $\log(D)$) | 97.5 | ±0.8 | 0.294 | ±.002 | 0.532 | ±.001 | 0.256 | ±.001 | 11.0 | ±0.1 |
| $GAN_2$(SCST, Co-att, $\log(D)$+5×CIDEr) | 111.1 | ±0.7 | 0.330 | ±.004 | 0.555 | ±.002 | **0.271** | ±.002 | 7.3 | ±0.2 |
| $GAN_3$(SCST, Joint-Emb, $\log(D)$) | 97.1 | ±1.2 | 0.287 | ±.005 | 0.530 | ±.002 | 0.256 | ±.002 | 11.2 | ±0.1 |
| $GAN_4$(SCST, Joint-Emb, $\log(D)$+5×CIDEr) | 108.2 | ±4.9 | 0.325 | ±.017 | 0.551 | ±.008 | 0.267 | ±.004 | 8.3 | ±1.6 |
| $GAN_5$(Gumbel Soft, Co-att, $\log(D)$) | 93.6 | ±3.3 | 0.282 | ±.015 | 0.524 | ±.007 | 0.253 | ±.007 | 11.1 | ±1.2 |
| $GAN_6$(Gumbel ST, Co-att, $\log(D)$) | 95.4 | ±1.5 | 0.298 | ±.009 | 0.531 | ±.005 | 0.249 | ±.004 | 10.1 | ±0.9 |
| $GAN_7$(Gumbel ST, Co-att, $\log(D)$+FM) | 92.1 | ±5.4 | 0.289 | ±.020 | 0.523 | ±.015 | 0.243 | ±.011 | 8.6 | ±0.8 |
| G-GAN Dai et al. (2017) from Table 1 | 79.5 | | 0.207 | | 0.475 | | 0.224 | | – | |
| CE* – * for non-attentional models | 87.6 | ±1.2 | 0.275 | ±.003 | 0.516 | ±.003 | 0.242 | ±.001 | 9.9 | ±0.8 |
| CIDEr-RL* | 100.4 | ±7.9 | 0.305 | ±.018 | 0.536 | ±.010 | 0.253 | ±.006 | 6.8 | ±1.4 |
| $GAN_1$*(SCST, Co-att, $\log(D)$) | 89.7 | ±0.9 | 0.276 | ±.000 | 0.518 | ±.001 | 0.246 | ±.001 | **13.2** | ±0.2 |
| $GAN_2$*(SCST, Co-att, $\log(D)$ + 5×CIDEr) | 103.1 | ±0.5 | 0.311 | ±.003 | 0.542 | ±.001 | 0.261 | ±.001 | 7.1 | ±0.2 |
| $GAN_3$*(SCST, Joint-Emb, $\log(D)$) | 90.7 | ±0.1 | 0.277 | ±.002 | 0.520 | ±.000 | 0.248 | ±.001 | 12.9 | ±0.1 |
| $GAN_4$*(SCST, Joint-Emb, $\log(D)$ + 5×CIDEr) | 102.7 | ±0.4 | 0.315 | ±.000 | 0.542 | ±.000 | 0.260 | ±.001 | 7.7 | ±0.1 |

improves the language metrics but sacrifices sentence diversity by reducing vocabulary coverage. Finally, note that the non-attentional models are behind in all metrics, except for vocabulary coverage. Interestingly, Co-att discriminators still provide better semantic scores than Joint-Emb despite non-attentional generators.

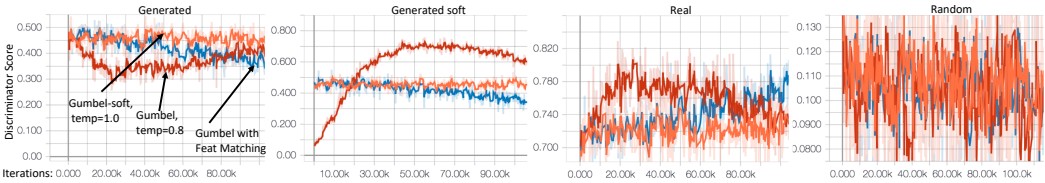

**Figure 4:** Discrimator scores during training across different Gumbel methods.

**SCST vs. Gumbel** Our experiments also showed that SCST is a more stable approach for training discrete GAN nodels, achieving better results as compared to Gumbel relaxation approaches.

To demonstrate that our experiments fairly compared both approaches, in Fig. 4 we show training of different Gumbel methods, where we plot the discriminator scores across gradient updates. As can be seen, at the end of training the generated sentences are scored around 0.5, random near 0.1 and real sentences above 0.7, indicating a properly trained discriminator and a healthy execution of all the Gumbel methods. Fig. 5 also compares gradient behaviors during training for SCST and Gumbel, showing that SCST gradients have smaller average norm and variance across minibatches, confirming our conclusion.

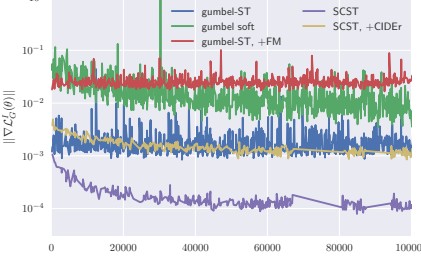

**Figure 5:** Training of $G_\theta$.

## 4 CONCLUSION

In summary, we demonstrated that SCST training for discrete GAN is a promising new approach that outperforms the Gumbel relaxation in terms of training stability and the overall performance. Moreover, we showed that our context-aware attention gives larger gains as compared to the adaptive sentinel or the traditional visual attention. Finally, our co-attention model for discriminator compares favorably against the joint embedding architecture.

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
