# OpenReview forum: "Improved Adversarial Image Captioning"
_ICLR.cc/2019/Workshop/DeepGenStruct — DeepGenStruct 2019_

### Official Review · AnonReviewer2 · 2019-04-12
**The experiments look solid, but the overall contribution of the paper looks marginal**

**Rating:** 3
**Confidence:** 3

**Review:**

The main contribution of this paper is an improved GAN model for image captioning. First, a context-aware LSTM captioner is proposed, which provides some moderate modifications to the original adaptive attention paper. Second, a stronger co-attentive discriminator is introduced, which shows better performance than previous discriminator design. Third, SCST is used for this GAN training.

Pros: The experiments are relatively well designed to understand the effect of each individual model design.

Cons:
The novelty of this paper is limited. It improves the original conditional GAN for image captioning marginally by using different generator and discriminator design, and a new training method. However, each individual module only provides marginal contributions. There is no surprise in the generator and discriminator design, and the usage of SCST for GAN training is also a direct application of previous methods.

---

### Official Review · AnonReviewer1 · 2019-04-15

**Rating:** 2
**Confidence:** 2

**Review:**

This paper compares two adversarial training approach for image captioning: self-critical sequence training (SCST) and Gumbel Straight-Through method. The discriminator utilizes a co-attention pooling mechanism to compute the compatibility of the caption and image. During training, the discriminator is trained to distinguish real captions from fake, as well as to detect unrelated real sentences (randomly chosen). To backpropogate the gradient from the discriminator to generator, the author using both SCST and Gumbel ST. The experimental results show that SCST performs better in terms CIDEr and BLEU.

This paper presents a thorough empirical evaluation between the two methods. However, the reviewer's main criticism of this paper is lack of technical innovation. Both SCST and Gumbel ST method have been proposed in previous work and the contribution of this paper is no more than empirical comparison.

---

### Decision · Program_Chairs · 2019-04-19
**Acceptance Decision**

**Decision:**

Accept

**Comment:**

As the reviewers note, it is true that both SCST and Gumbel ST have been utilized in the past for reinforcement-learning style problems. However the application to GAN-based image captioning and the thorough experiments will make a nice contribution to the workshop.